# Molecular Screening of Bioactive Compounds of Garlic for Therapeutic Effects against COVID-19

**DOI:** 10.3390/biomedicines11020643

**Published:** 2023-02-20

**Authors:** Huma Ashraf, Erum Dilshad, Tayyaba Afsar, Ali Almajwal, Huma Shafique, Suhail Razak

**Affiliations:** 1Department of Bioinformatics and Biosciences, Faculty of Health and Life Sciences, Capital University of Science and Technology (CUST), Islamabad 44000, Pakistan; 2Department of Community Health Sciences, College of Applied Medical Sciences, King Saud University, Riyadh 11433, Saudi Arabia; 3Institute of Cellular Medicine, Newcastle University Medical School, Newcastle University, Newcastle NE1 7RU, UK

**Keywords:** SARS-CoV-2, *Allium sativum*, molecular docking, lead compounds, Allicin, Diallyl Sulfide, Diallyl Disulfide, Diallyl Trisulfide, Ajoene, Levamisole

## Abstract

An outbreak of pneumonia occurred on December 2019 in Wuhan, China, which caused a serious public health emergency by spreading around the globe. Globally, natural products are being focused on more than synthetic ones. So, keeping that in view, the current study was conducted to discover potential antiviral compounds from *Allium sativum*. Twenty-five phytocompounds of this plant were selected from the literature and databases including 3-(Allylsulphinyl)-L-alanine, Allicin, Diallyl sulfide, Diallyl disulfide, Diallyl trisulfide, Glutathione, L-Cysteine, S-allyl-mercapto-glutathione, Quercetin, Myricetin, Thiocysteine, Gamma-glutamyl-Lcysteine, Gamma-glutamylallyl-cysteine, Fructan, Lauricacid, Linoleicacid, Allixin, Ajoene, Diazinon Kaempferol, Levamisole, Caffeicacid, Ethyl linoleate, Scutellarein, and S-allylcysteine methyl-ester. Virtual screening of these selected ligands was carried out against drug target 3CL protease by CB-dock. Pharmacokinetic and pharmacodynamic properties defined the final destiny of compounds as drug or non-drug molecules. The best five compounds screened were Allicin, Diallyl Sulfide, Diallyl Disulfide, Diallyl Trisulfide, Ajoene, and Levamisole, which showed themselves as hit compounds. Further refining by screening filters represented Levamisole as a lead compound. All the interaction visualization analysis studies were performed using the PyMol molecular visualization tool and LigPlot+. Conclusively, Levamisole was screened as a likely antiviral compound which might be a drug candidate to treat SARS-CoV-2 in the future. Nevertheless, further research needs to be carried out to study their potential medicinal use.

## 1. Introduction

An outbreak of pneumonia pandemic occurred in China, in December 2019, which caused a serious public health emergency by spreading around the globe. Finally, it was officially announced on 9 January 2020 that the outbreak in Wuhan is caused by a novel coronavirus 2019-nCoV. On 11 March, COVID-19 was declared a pandemic disease by the WHO, as it is easily transferable from one to another human. Globally, nearly 1.9 million new cases and over 12,000 deaths were reported in the week of 16 to 22 January 2023 [1]. The unique coronavirus was named Severe Respiratory Disease [2]. Coronaviruses (CoVs) are responsible for causing infection in humans as well as in animals, and they cause several other diseases related to respiratory issues. These spreader viruses are grouped into alpha, beta, gamma, and delta variants, with a new Omicron variant appearing recently [3]. 

The pandemic which occurred in the period 2002–2003, in China and Asia Pacific regions, was caused by SARS-CoV and infected more than 8000 people around the globe, with a 10% mortality rate. Fever, cough, and lowering of oxygen level in blood were the common symptoms that were seen in patients suffering from the illness [4]. The sequence similarity of the SARS-CoV-2 virus is nearly 80% when compared with the SARS-CoV virus, but the coronavirus is much more severe and dangerous [5]. Whole-genome sequencing revealed that coronavirus is more related to bat CoV RaTGI3, with a 96.2% sequence similarity. 

The virus transmits by contact in any form from one to another human being, i.e., direct or indirect contact [6]. Approximately 2–14 days is the incubation period of the virus. Moreover, few infected people are asymptomatic, which means no symptoms of the disease are shown. COVID-19 is mostly not so severe; sometimes, patients experience health issues such as hypertension, diabetes, immunodeficiency, etc. For such patients, multi-organ failure may occur in case of severe conditions which can cause death [4]. 

Globally, for the treatment of COVID-19, without having proof of inflammation’s role in the cure of the illness, many immune modulators, such as glucocorticoids and anti-inflammatory therapies are being used for this purpose. The major determinant in the host’s survival is dependent on the host’s ability to clear the viral infection in the lung, which provides an advantage to the host by aiding effective viral clearance [7]. The pathophysiological mechanism of COVID-19 is not well understood, and several pieces of evidence have revealed that COVID-19-infected patients have high levels of cytokine and are referred to as cytokine storm or cytokine release syndrome. This abnormal rise in cytokine level is considered a severe decline in health conditions in the infected patients. Thus, the severity of disease in COVID-19-infected patients can be reduced by suppressing elevated inflammatory response [8]. 

Effective measures, such as vaccines [9], small-molecule inhibitors [10], and bioactive natural products [11,12], are greatly needed to reduce SARS-CoV-2 transmission. However, promising drugs for SARS-CoV-2 treatment do not exist [11]. As a key component of the COVID-19 treatment regimen, transactional medicine [13], including garlic, may demonstrate potential value in countering SARS-CoV-2 infection. This is because the secondary metabolites that are present in medicinal plants can prevent viral penetration and replication by binding with viral proteins and enzymes. Plants contain a lot of bio-active compounds and essential oils which are beneficial for human health. Another factor that increased the demand for chemical-free herbal drugs was the toxic and adverse effects of allopathic medicines. The problem is that the distribution of medicinal plants is not the same worldwide, and usually, medicinal herbs are collected from wildlife populations [14]. Moreover, the demand for wildlife resources in Europe, North America, and Asia has increased from 8 to 15% per year. Naturally occurring spices and their isolated active components have been reported for targeting anti-inflammatory pathways, inducing anti-inflammatory effects in several life-threatening ailments. Spices and herbs are thought to be excellent immunity boosters; therefore, they are prevalently used in Asian countries [15].

The viral main proteinase (Mpro, also called 3CLpro), which controls the activities of the coronavirus replication complex, is an attractive target for therapy [16]. The objective of the current study was to screen the bioactive compounds of the *Allium sativum* plant, effective against COVID-19 by determining their binding confirmation with the target protein (3CL protease) of COVID-19. An additional aim was to study the interaction between targeted proteins and the selected ligands computationally.

## 2. Materials and Methods

### 2.1. Selection of Protein

3CL protease of COVID-19 was selected as the target protein for the current study. The structure of the SARS-CoV-2 selected protein was retrieved from Protein Data Bank (PDB ID: 6M2Q) in .pdb format [16].

### 2.2. Primary Sequence Retrieval

The primary sequence of target proteins was taken in FASTA format from the protein sequence database UniProt (http://www.uniprot.org accessed on 12 February 2023 (PRJNA318322)) [17].

### 2.3. Analysis of Physiochemical Properties

This was to determine the function of proteins. ProtParam was used to predict the physicochemical parameters of SARS-CoV-2 protein including molecular weight, number of amino acids, isoelectric point, instability index, and grand average of hydropathicity (GRAVY). ProtParam tool of ExPASy was used to determine the negatively charged residues (Asp + Glu), positively charged residues (Arg + Lys), aliphatic index, and atomic composition [18].

### 2.4. Identification of Functional Domains

Interpro (https://www.ebi.ac.uk/interpro/D1/D344/5958491 accessed on 12 February 2023) was used to detect and predict the functional domain of targeted protein. Conserved domains are involved in sequence/structure/relationship study. InterPro provides a practical analysis of proteins by classifying them into families and predicting domains and active sites [19]. 

### 2.5. Active Site Identification

The ligand shows maximum or highest interaction with the protein where the target protein has its active site. Amino acids are highly involved in the formation of a complex of ligands to protein. Protein binding pockets were identified by CASTp [19].

### 2.6. Ligand Preparation

The 3-dimensional (3D) structure of ligands was obtained from PubChem. PubChem is the world’s largest collection of freely available chemical information. We can search several ligands by their names, molecular formula, and structure and by other information. If the targeted structure is not available in PubChem, then can be drawn via ChemDraw by inserting Canonical smileys derived from PubChem [16].

### 2.7. Bioactivity Analysis of Ligands and Toxicity Measurement

Selected ligands from the PubChem database should follow the Lipinski rule of five, having required chemical and physical properties. This was performed by using PkCSM (https://omictools.com/pkcsmtool/database/id/1618 accessed on 12 February 2023) [18].

### 2.8. Molecular Docking Process

The purpose of molecular docking is to find the best conformational interaction between target proteins and compounds. Molecular docking of protein and ligands was performed through Cavity detection-guided Blind Docking (CB-Dock) [20].

### 2.9. Visualization of Ligand/Protein 

Docked complex of ligands and protein was visualized by PyMol. Docking poses generated via CB-Dock were visualized and saved as a molecule in .pdb form in one file for further analysis [18].

### 2.10. Analysis of Docked Complex 

Analysis of docked complex was performed by using LigPlot+, which automatically generates schematic diagrams of protein–ligand interactions for a given PDB file. These interactions are modified by hydrogen bonds and through hydrophobic contact [18].

### 2.11. Ligand ADMET Properties

The main aim of predicting ADMET is to choose strong candidates by eliminating weak drug candidates in the early stages of drug development. Optimization of the ADMET (Absorption, Distribution, Metabolism, Excretion, and Toxicity) properties of the drug molecule was performed by using PkCSM [21]. 

### 2.12. Active Inhibitor Identification

After a detailed analysis of protein and ligand interactions, docking scores, and toxicity studies, the most active inhibitor was identified. The selected compound was our lead compound [19].

### 2.13. FDA-Approved Drug-Proposed Antiviral Agent Comparison

Finally, the comparison was made between the selected antiviral drug Remdesivir and the proposed antiviral agents by comparing all the parameters described above [22].

## 3. Results and Discussion

### 3.1. Target Proteins Structure and Properties

The primary sequence of the target protein (3CL Protease) was taken in FASTA format from the UniProt database under accession numbers P0DTD1 with 7096 residues length. The 3D structure of 3CLpro of SARS-CoV-2 was obtained from Protein Data Bank (PDB ID: 6M2Q) in .pdb format [16]. Physiochemical properties of 3CL protease were determined by ProtParam under accession No. [A0A6C0M8P6-SARS2]. In physicochemical parameters of selected protein 3CLpro of SARS-CoV-2, Mol. weight, atomic composition, isoelectric point, no. of amino acids, instability index, grand average of hydropathicity (GRAVY), No. of negatively charged residues (Asp + Glu), No. of positively charged residues (Arg + Lys), Aliphatic index, and amino acid and atomic composition were included, and these properties were investigated using the ProtParam ExPASy tool. InterPro (https://www.ebi.ac.uk/interpro/14231 accessed on 12 February 2023) was used to identify active domains of 3CLpro domains [16]. The selected target protein is shown in Figure 1. 

SARS-CoV-2 is the virus which is responsible to cause COVID-19 and up till now there is no proper treatment for this pandemic which has affected the world. To know about the virus, it is compulsory to obtain information about the structure of the involved virus. So, this structure (Figure 1) can be understood in a better way. Accordingly, 3CL protease comes from the class of highly conserved viruses, and it is now the target of broad-spectrum antiviral drugs which kill the virus as it is the site of replication of the virus [23]. In the early studies on the SARS-CoV-2 models, Mpro shows a close relation to the main proteases named coronaviral in terms of structure: 99% of the amino acid structure is common to bat CoV RaTG13 Mpro and 97% similar to SARS CoV Mpro [24]. 

The functional analysis of protein sequences was obtained by Interpro to determine the conserved and functional domain and sequence/structure/relationship. More than one functional domain can be there to perform different functions [19]. These conserved domains and families of the target protein are shown in Figure 2. 

Figure 3 shows functional domains and pockets present in red color along with the structure of the protein. Moreover, Table 1 shows the area and volume of these pockets which were obtained by using CASTp software.

### 3.2. Ligand Selection and Molecular Docking

The 3D structures and information of selected ligands that are 3-(Allylsulphinyl)- L-alanine, Allicin, Diallyl sulfide, Diallyl disulfide, Diallyl trisulfide, Glutathione, L-Cysteine, S-allyl-mercapto-glutathione, Thiocysteine, gamma-glutamyl-Lcysteine, gamma-glutamyl allyl cysteine, Quercetin, myricetin Kaempferol, Fructan, Lauric acid, Linoleic acid, Allixin, Ajoene, diazinon, levamisole, caffeinated, Ethyl linoleate, Scutellarein, and S-allyl cysteine methyl ester were downloaded from PubChem in SDF format [25].

The 3D structure of the target proteins and the ligands was taken as the input for docking, which was performed by the CB dock [21]. The CB dock gave possible poses with receptor models, and among these poses, the best one was selected by observing certain properties such as vena score, size of cavity, etc. [19]. The CB Dock also projected the predictable binding site for protein and premeditated centers and sizes with an innovative rotation cavity detection method and performed docking with the popular docking program known as Auto dock Vina [19]. So, the obtained data is given in Table 2, which shows the minimum and maximum energy, cavity size, binding score, and grid map of ligands.

LigPlot+ (version v.1.4.5) and PyMol Edu (v1.7.4.5) were used for analyzing docking results. LigPlot+ (version v.1.4.5) also determined the interactions of ligands and target proteins [26]. The graphical system of LigPlot+ automatically generates multiple 2D diagrams of interactions from 3D coordinates. The 2D diagrams of the best binding score ligands with respective proteins were obtained from LigPlot+, shown in Figure 4A–P. As evident from the 2D diagram, ligands show only hydrophobic interactions with the protein.

The ligand consisted of 10 carbons and showed hydrophobic interactions with Pro132, Pro293, Pro108, Thr292, Gly109, Ile200, Ile249, Glu240, His246, Val202, and Phe294 residues, and it included Allicin, Diallyl Sulfide, Diallyl Disulfide, Diallyl Trisulfide, levamisole, diazinon, thiocysteine, gamma-glutatmylS-allyl cysteine, and ajoene. These ligands were without hydrogen bonds, as it is evident from the 2D structures they are mostly without active oxygen atoms. S-allyl cysteine methyl ester, ethyl linoleate, and linoleic acid had one hydrogen bond. S-allylmercapto-glutathiole, caffeic acid, scutellarein, allixin, lauric acid, and quercetin had two hydrogen bonds. L-cystein had 3 hydrogen bonds, whereas 3-L-alanine, kaempferol, and glutathione had 4 hydrogen bonds. Maximum hydrogen bonds are shown by fructan, myricetin, and gamma-glutamyl-L-cysteine as five hydrogen bonds each [27,28].

### 3.3. ADMET Properties of Ligands

Lipinski’s five drug laws, when applied, served as a first filter in assessing the drug likelihood of the selected ligands. In our thesis, 25 different ligands were taken, and when filtered by different software, few were left. So, when Lipinski’s rule of five was applied, Myricetin, Fructan, Linoleic Acid, Ethyl Linoleate, Glutathione, and S-Allyl-Mercapto-Glutathione were knocked out as shown in Table 3. 

Further, ligands were screened by calculating the ADMET (absorption, distribution, metabolism, excretion, and toxicity) properties as a measure of pharmacokinetics using the online tool PkCSM [29]. ADMET properties are shown in Table 4, Table 5, Table 6, Table 7 and Table 8, respectively. 

## 4. Lead Compounds Identification

Myricetin, Fructan, Linoleic Acid, Ethyl Linoleate, Glutathione, and S-Allyl-Mercapto-Glutathione were knocked out from Lipinski’s rule of five. 3-(Allylsulphinyl)-L-Alanine, Scutellarein, Diazinon, Glutathione, L-Cysteine, S-Allyl-Mercapto-Glutathione, Thiocysteine, Kaempferol, Quercetin, Myricetin Fructan, Lauric Acid, Linoleic Acid, Allixin, Gamma-Glutamyl-L-Cysteine, Gamma-Glutamyl-S-Allylcysteine, S-allylcysteine methylester, and Caffeic Acid had a logBB value > 0.3. 3-(Allylsulphinyl)-L-Alanine, Scutellarein, Diazinon, Glutathione, L-Cysteine, Gamma-Glutamyl-S-Allylcysteine, S-allylcysteine methylester, Caffeic Acid, S-Allyl-Mercapto-Glutathione, Thiocysteine, Gamma-Glutamyl-L-Cysteine, Kaempferol, Quercetin, Myricetin Fructan, Lauric Acid, Linoleic Acid, Allixin, and Ethyl Linoleate had a logBB value > 0.3 and logPS value > −2.

Linoleic Acid and Ethyl Linoleate have a logPS value > −2. So, the lig can be identified as lead compounds. The best five compounds were Allicin, Diallyl Sulfide, Diallyl Disulfide, Diallyl Trisulfide, Ajoene, and Levamisole. The lead compound of this research work was Levamisole, as is also indicated by molecular docking [20].

## 5. Drug Identification against COVID-19

With the emergence of the disease, many FDA-approved drugs were utilized for drug repurposing, finding the best treatment against the virus. One of the drugs that has been in use in different countries such as the UK, Brazil, India, Pakistan, and many more is Remdesivir. Though the use of this medicine has been increased during this whole pandemic, this drug is still in clinical trials [22]. The first FDA-approved drug to treat SARS-CoV-2 is Remdesivir, which is an antiviral nucleotide analogue prodrug [22]. Because of its broad-spectrum nature and mechanism of action against various viral families, it is suggested to the patients. This medicine is a non-obligate chain terminator of RdRp from SARS-CoV-2 and the related SARS-CoV and MERS-CoV, and it has been investigated and suggested in many different clinical trials against COVID-19 [30].

## 6. Reference Drug ADMET Properties

The drug ADMET properties were studied by using the same software as above, which is PkCSM.

### 6.1. Absorption Properties

Table 9 shows the absorption properties of Remdesivir. The values show that Remdesivir shows a very low CaCO2 solubility and water solubility. Though intestinal absorption is high, it is still in the safe range. Remdesivir also has a lower value of skin permeability. Remdesivir is also a P-glycoprotein substrate and an inhibitor of P-glycoprotein I but not a P-glycoprotein II inhibitor.

### 6.2. Distribution Properties

Table 10 shows the distribution properties of Remdesivir. The distribution parameters value shows that the value of VDss is low, which means the drug would not be distributed properly. Remdesivir can penetrate in CNS and also can pass the blood–brain barrier.

### 6.3. Metabolic Properties

Table 11 shows the metabolic properties of Remdesivir. It indicates that Remdesivir is not a CYP2D6 substrate; rather, it is a CYP3A4 substrate. With those, Table 11 shows that Remdesivir is not a CYP1A2, CYP2C19, CYP2C9, CYP2D6, and CYP3A4 inhibitor.

### 6.4. Excretion Properties

Table 12 shows the excretion properties of Remdesivir. The above table gives the values of the Excretory properties of Remdesivir. It shows that Remdesivir is not a renal OCT2 substrate, which means it will not help in clearing the drug. With that, the value of total clearance as 0.198 is also given with respect to its liver.

### 6.5. Toxicity Prediction of Reference Drug

Table 13 shows the Toxicity Properties of Remdesivir. The toxicity parameters value of Remdesivir shows that this drug can be toxic towards the liver, but other parameters are in the range of positive values. 

This indicates that Remdesivir can cause any skin sensitivity, and it also is not an inhibitor of hERG I but an hERG II inhibitor. The dose value of 0.291 is also tolerable. With that, a no to AMES toxicity indicates that it is not carcinogenic.

## 7. Remdesivir Molecular Docking

Table 14 shows the docking result of Remdesivir. The table indicates that Remdesivir has a binding score of −8.1. The docking results of Remdesivir show that it has quite a good binding score. Additionally, has four hydrogen bond donors and thirteen hydrogen bond acceptors that break two of Lipinski’s rules, as the molecular weight is above 500 g/mol.

## 8. Remdesivir Comparison with Lead Compound

The standard drug Remdesivir was compared with the lead compound Levamisole and its physicochemical and pharmacokinetic properties. Table 15 shows that Remdesivir breaks two of Lipinski’s rules relating to molecular weight and H-bond acceptor: the molecular weight of Remdesivir is 602.585, which is greater than the 500 allowed according to Lipinski, and the H-bond acceptor of Remdesivir accepts 13 hydrogens, but according to Lipinski, it should not be more than 10; in contrast, Levamisole follows all rules of LogP, Molecular weight, H-bond acceptor, and H bond donor according to Lipinski.

## 9. ADMET Properties Comparison

The ADMET properties comparison was performed to check the absorption, distribution, metabolic excretion, and toxicity properties of the drug and the lead compound, in order to find a better drug candidate.

### 9.1. Absorption Properties Comparison

The parameter of absorption is based on 6 models. The water solubility model gives the value of the compound’s solubility in the water at 25‰. A model of CaCO2 solubility is used to detect the absorption of the drug. Values greater than 0.90 are considered to have high intestinal absorption, which means Levamisole is absorbed more than Remdesivir. The value of the intestinal absorption model is less than 30%, which means the drug is not well absorbed. The given values of both the standard and lead compound show that Levamisole has high intestinal absorption. 

For the transdermal drugs skin permeability model, a value less than log Kp > −2.5 is considered low; according to this, both compounds pass the skin permeability test. The P-glycoprotein substrate model is very important, as P-glycoprotein is an ABC transporter. Both Levamisole and Remdesivir act as substrates. The last model of P-glycoprotein inhibitors shows whether the compound is an inhibitor or not [31]. Table 16 shows that Levamisole is an inhibitor of P-glycoprotein II, whereas Remdesivir is the inhibitor of P-glycoprotein I.

### 9.2. Metabolic Properties Comparison

Cytochrome P450 is mainly found in the liver and is held responsible for oxidizing the xenobiotic so that they can be excreted easily out from the body, hence making cytochrome P450 a detoxification enzyme. Some drugs are activated by it, and some are deactivated [32]. Table 17 shows that Remdesivir is a CYP3A4 substrate, and Levamisole is a CYP3A4 substrate and CYP2D6 inhibitor.

### 9.3. Distribution Properties Comparison

Table 18 shows the comparative distribution properties of Remdesivir and Levamisole. The distribution parameter is based on 4 models. The volume of distribution (VDss) is a uniform distribution of the drug in the blood plasma, and if this value is above 2.81 L/kg, then the drug is distributed more in the tissues rather than in the blood plasma. Both Remdesivir and Levamisole have a reasonable VDss value. The 2nd model is based upon the unbound fraction of the drugs in the plasma, as bounded drugs affect the efficiency of the drugs. The given value is the amount of the drug which remains unbounded [33]. For BBB permeability, if the value is greater than 0.3 logBB, then that drug can easily cross the blood–brain barriers, and if the value is less than −1 logBB, then the drug does not properly reach the brain [34]. By these values, it is clear that Remdesivir has a low value; hence, it would be poorly distributed to the brain. Similarly, the model for CNS is based on the values that if the logPS > −2, then that drug can easily penetrate the CNS, while those having value of logPS < −3 are unable to reach the CNS. Remdesivir has a low value, and hence, it will not cross and reach the CNS.

### 9.4. Excretion Properties Comparison

Levamisole has more total clearance than Remdesivir. The 2nd model is of the Renal OCT2 (organic cation transporter 2), and this transporter helps in renal clearance. Being an OCT2 substrate, it can show an adverse effect in correlation with inhibitors [35]. So, both Remdesivir and Levamisole are not Renal OCT2 substrates. Table 19 shows the values of excretory properties of Remdesivir and Levamisole.

### 9.5. Toxicity Comparison

The toxicity of both the standard drug and lead compound was based on nine models. Model 1 of AMES toxicity shows that both the standard and lead compounds are not mutagenic. Model 2 of the maximum tolerated dose shows that if the value is equal to or less than 0.477 log mg/kg/day, then it is considered low, and greater values are considered high. 

Table 15 below shows that Levamisole has a low value of the tolerated dose. The 3rd model is of hERG I and II inhibitors, where only levamisole is an inhibitor of both, while remdesivir inhibits only II inhibitors. The 4th model of oral rat acute toxicity is used to assess the relative toxicity. Model 5 of oral rat chronic toxicity gives the values of the lowest dose that could result in an adverse effect [34]. 

Model 6 of hepatotoxicity shows that the drug can cause damage to the liver. Table 20 shows that Remdesivir is hepatotoxic. For the dermal products model, Model 7 is used for checking the sensitivity towards the skin. Both the standard and lead compounds are not sensitive to skin. Model 8 uses *T. pyriformis*, and Model 9 uses minnows to check the toxicity [36].

For *T. pyriformis*, a value > −0.5 is considered toxic, according to which Remdesivir is somewhat toxic, and minnow toxicity values below 0.5mM are considered toxic, and both compounds pass this toxicity test. Table 20 shows the comparative values of toxicity of Remdesivir and Levamisole.

### 9.6. Physiochemical Properties Comparison

For determining the fundamental properties of the compounds, physiochemical properties were studied. This screening shows that Remdesivir has 27 carbon atoms, 35 hydrogen atoms, 6 nitrogen atoms, 8 oxygen atoms, and a phosphorous atom, whereas Levamisole has 11 carbon atoms, 12 hydrogen atoms, 2 nitrogen atoms, and Sulphur. Remdesivir can donate 4 hydrogen atoms, whereas Levamisole cannot donate hydrogen. 

Remdesivir can accept 13 Hydrogen atoms which do not fall under the Lipinski rule. Although the Log P value of Remdesivir is more than that of Levamisole, the molecular weight of Remdesivir is far greater than Levamisole, and also, it does not fall under the Lipinski rule. Table 21 shows the comparison of the physiochemical properties of Remdesivir and Levamisole.

### 9.7. Docking Score Comparison

Both the standard and the lead compound were docked, and the docking result gives us the best binding score. Table 22 shows that the lead compound Levamisole, which has a much higher Vina score than that of the standard drug, which is Remdesivir. The binding score of Remdesivir is −8.1 and that for Levamisole is −5.7, which is higher. This result shows that Levamisole can block the 3CL pro or bind with it more efficiently than can Remdesivir.

## 10. Conclusions

The motive of the present research was to discover potential antiviral components from *Allium sativum*. Twenty-five phytocompounds (which represent almost all classes of natural antiviral compounds) were selected from the literature and databases. Molecular docking was performed by CB-dock, an online tool against 3CL protease of COVID-19 and the five best-scoring phytocompounds were identified as hit compounds. Physicochemical and pharmacokinetic properties determined the final destiny of compounds as drug or non-drug compounds. Levamisole was predicted as a lead compound by virtual screening results. As per the results of this research, the lead compound, Levamisole, can be explored as an important candidate to cure viral infections, especially COVID-19. These potential antiviral compounds of *Allium sativum* can also be tested for the pharmaceutical and medical industries. 

## Figures and Tables

**Figure 1 biomedicines-11-00643-f001:**
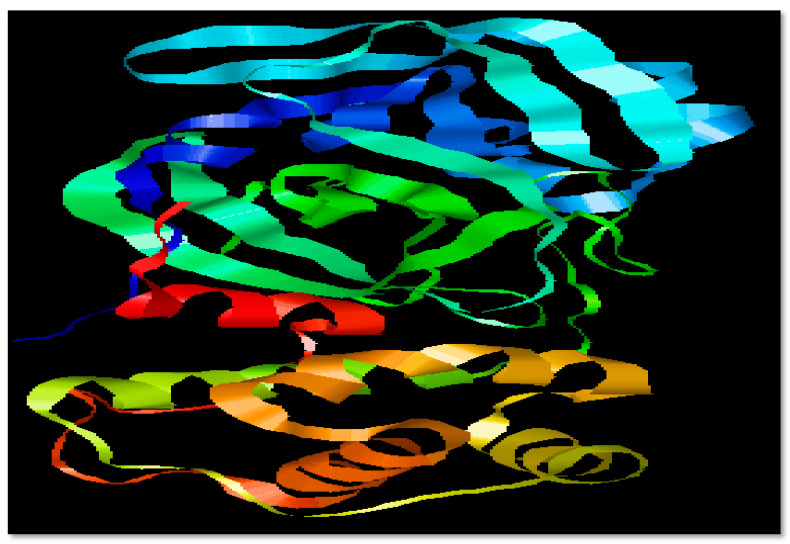
Structure of SARS-CoV-2 3CL protease (3CL pro).

**Figure 2 biomedicines-11-00643-f002:**
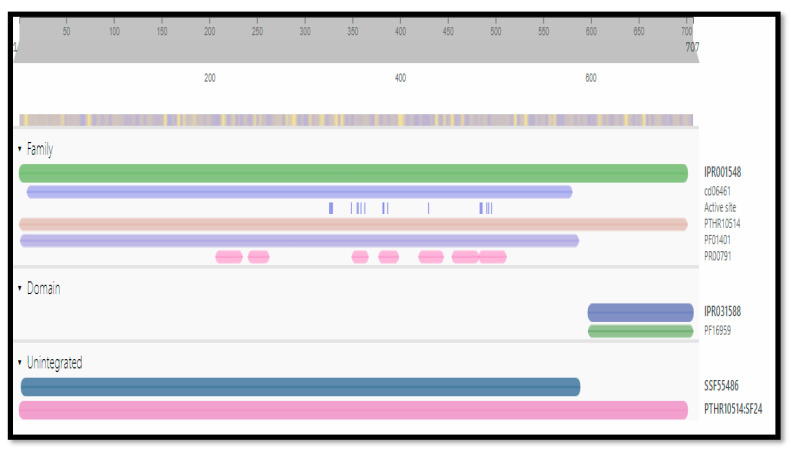
Conserved domains of the target protein 3CL protease.

**Figure 3 biomedicines-11-00643-f003:**
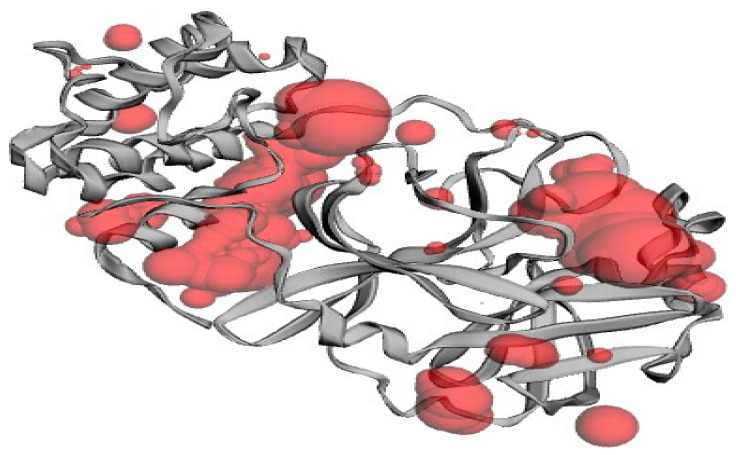
Functional domains of the target protein.

**Figure 4 biomedicines-11-00643-f004:**
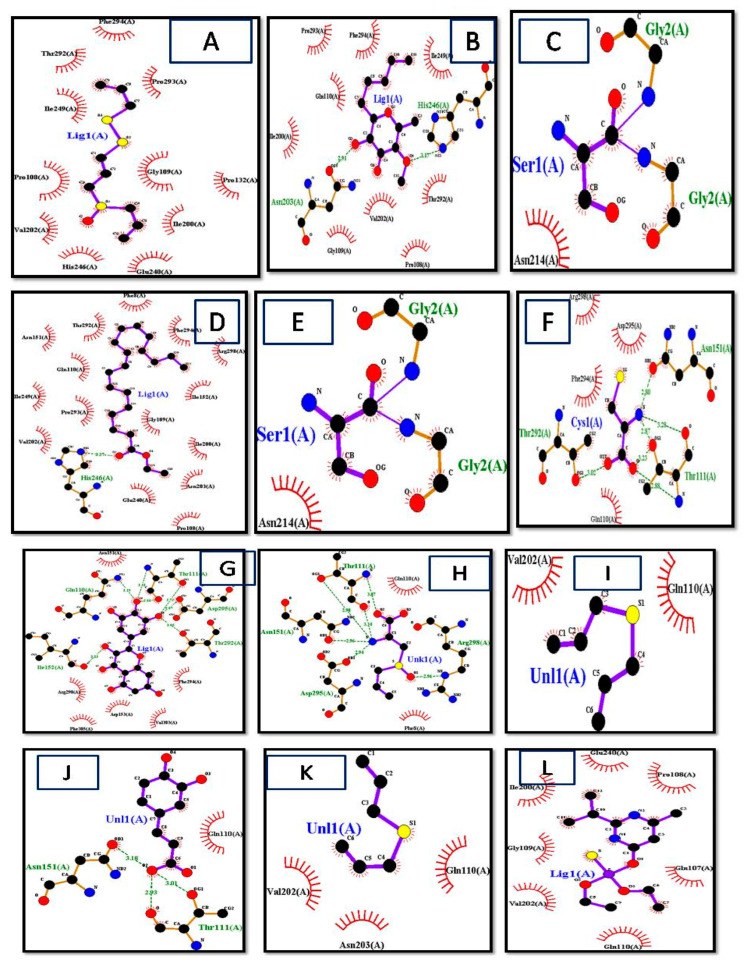
Interactions of (**A**) Ajoen, (**B**) Allixin, (**C**) diallyl disulfide, (**D**) linoleate, (**E**) Gamma-Glutamyl-S-Allylcysteine, (**F**) lauric acid, (**G**) l-cystein, (**H**) Myrecetin, (**I**) 3-(Allylsulphinyl)-L-Alanine, (**J**) allicin, (**K**) caffeic acid, (**L**) diallyl sulfide, (**M**) diazinon, (**N**) fructan, (**O**) Gamma-Glutamyl-L-Cysteine, (**P**) Lauric acid, (**Q**) Levimisole, (**R**) Linoleic acid, (**S**) quercetin, (**T**) S-allylcysteine, (**U**) S-allyl-mercapto-glutathione, (**V**) scutellarein, (**W**) glutathonine, and (**X**) f diallyl trisulfide with target protein obtained by LigPlot+.

**Table 1 biomedicines-11-00643-t001:** Area and volume of the obtained pockets by CASTp.

Pocket ID	Area (SA)	Volume (SA)
1	284.664	292.690
2	273.913	214.993
3	53.189	59.074
4	104.306	30.390
5	40.514	25.655
6	27.458	7.991
7	20.634	7.083
8	13.671	4.642
9	6.817	3.399
10	16.544	3.174
11	10.896	2.007
12	10.676	1.878
13	6.664	1.663
14	6.528	0.747
15	4.217	0.743
16	6.818	0.470
17	4.637	0.424
18	4.031	0.356
19	1.004	0.163
20	1.770	0.100
21	0.993	0.038
22	0.744	0.020
23	0.265	0.006
24	0.103	0.002
25	0.071	0.001
26	0.046	0.000
27	0.001	0.000
28	0.022	0.000
29	0.000	0.000
30	0.013	0.000
31	0.042	0.000

**Table 2 biomedicines-11-00643-t002:** Results of CB dock of selected ligands with the target protein.

Sr. No	Ligand	Binding Score	Cavity Size	Grid Map	Min-Energy (Kcl/mol)	Max-Energy (Kcl/mol)
**1**	**3-(Allylsulphinyl)-L-Alanine**	−4.8	1385	26	0	1.6 × 10^0^
**2**	**Allicin**	−3.2	1385	26	0	1.6 × 10^0^
**3**	**Diallyl Sulfide**	−3.1	1385	26	0	1.6 × 10^0^
**4**	**Diallyl Disulfide**	−3.5	1385	26	0	1.6 × 10^0^
**5**	**Diallyl Trisulfide**	−5.5	277	21	0	1.6 × 10^0^
**6**	**Glutathione**	−5.8	277	21	0	1.6 × 10^0^
**7**	**L-Cysteine**	−3.7	1385	26	0	1.6 × 10^0^
**8**	**S-Allyl-Mercapto-Glutathione**	−6	277	22	0	1.6 × 10^0^
**9**	**Thiocysteine**	−3.8	1385	21	0	1.6 × 10^0^
**10**	**Gamma-Glutamyl-L-Cysteine**	−7.2	1385	24	0	1.6 × 10^0^
**11**	**Gamma-Glutamyl-S-Allylcysteine**	−5.4	1385	21	0	1.6 × 10^0^
**12**	**Kaempferol**	−7.4	1385	26	0	1.6 × 10^0^
**13**	**Quercetin**	−7.6	1385	21	0	1.6 × 10^0^
**14**	**Myricetin**	−7.8	1385	21	0	1.6 × 10^0^
**15**	**Fructan**	−7.1	1385	21	0	1.6 × 10^0^
**16**	**Lauric Acid**	−5.2	1385	22	0	1.6 × 10^0^
**17**	**Linoleic Acid**	−5.7	1385	30	0	1.6 × 10^0^
**18**	**Allixin**	−5.8	1385	26	0	1.6 × 10^0^
**19**	**Ajoene**	−4.7	1385	22	0	1.6 × 10^0^
**20**	**Ethyl Linoleate**	−5.8	1385	31	0	1.6 × 10^0^
**21**	**Diazinon**	−5.7	1385	26	0	1.6 × 10^0^
**22**	**Levamisole**	−5.7	1385	26	0	1.6 × 10^0^
**23**	**Scutellarein**	−7.6	1385	21	0	1.6 × 10^0^
**24**	**S-allyl cysteine methyl ester**	−5.3	1385	21	0	1.6 × 10^0^
**25**	**Caffeic acid**	−5.8	1385	26	0	1.6 × 10^0^

**Table 3 biomedicines-11-00643-t003:** Results of ligands under Lipinski’s rule of five.

Sr. No	Ligand	LogP	Molecular Weight(g/mol)	Hydrogen Bond Acceptor	Hydrogen Bond Donar
1	L-Alanine	−0.667	177.225	3	2
2	Allicin	1.7553	162.279	2	0
3	Diallyl Sulfide	2.0916	114.213	1	0
4	Diallyl Disulfide	2.7398	146.28	2	0
5	Diallyl Trisulfide	3.388	178.347	3	0
6	Glutathione	−2.2061	307.328	6	6
7	L-Cysteine	−0.6719	121.161	3	3
8	S-Allyl-Mercapto- Glutathione	−0.741	379.46	7	6
9	Thiocysteine	−0.0237	153.228	4	3
10	L-Cysteine	l- −0.0227	429.503	l- 10 10	6
11	Gamma-Glutamyl- - S-Allylcysteine	−3329	290.341	5	4
12	Kaempferol	2.2824	286.239	6	4
13	Quercetin	1.988	302.238	7	5
14	Myricetin	1.6936	318.237	8	6
15	Fructan	−7.5682	504.438	16	11
16	Lauric Acid	3.9919	200.322	1	1
17	Linoleic Acid	5.8845	280.452	1	1
18	Allixin	2.39512	226.272	4	1
19	Ajoene	3.0022	234.411	3	0
20	Ethyl Linoleate	6.363	308.506	2	0
21	Diazinon	3.58472	304.352	6	0
22	Levamisole	2.1461	204.298	3	0
23	Scutellarein	2.2824	286.239	6	4
24	S-allyl cysteine methyl ester	1.9719	275.37	5	1
25	Caffeic acid	2.987	180.159	3	3

**Table 4 biomedicines-11-00643-t004:** Absorption properties of ligands.

Sr. No	Ligand	Water Solubility	Caco2 Permeability	Intestinal Absorption (Human)	Skin Permeability	P-Glucoprotein Substrate	P-Glucoprotein I inhibitor	P-glucoprotein II Inhibitor
1	-L-Alanine	−2.888	0.619	76.495	−2.735	No	No	No
2	Allicin	−1.72	1.316	96.229	−1.877	No	No	No
3	Diallyl Sulfide	−2.695	1.394	96.268	−1.488	No	No	No
4	Diallyl Disulfide	−3.222	1.399	94.769	−1.429	No	No	No
5	Diallyl Trisulfide	−3.781	1.403	92.573	−1.449	No	No	No
6	Glutathione	−2.892	−0.536	0	−2.735	Yes	No	No
7	L-Cysteine	−2.888	0.386	74.807	−2.737	No	No	No
8	S-Allyl-Mercapto- Glutathione	−2.205	−0.457	0	−2.735	Yes	No	No
9	Thiocysteine	−2.887	0.424	78.653	−2.737	No	No	No
10	Gamma-Glutamyl- L-Cysteine	−2.892	−0.598	0.259	−2.735	Yes	No	No
11	Gamma-Glutamyl- S-Allylcysteine	−2.891	−0.517	8.312	−2.735	Yes	No	No
12	Kaempferol	−3.04	0.032	74.29	−2.735	Yes	No	No
13	Quercetin	−2.925	−0.229	77.207	−2.735	Yes	No	No
14	Myricetin	−2.915	0.095	65.93	−2.735	Yes	No	No
15	Fructan	−1.2	−0.835	0	−2.735	Yes	No	No
16	Lauric Acid	−4.181	1.562	93.379	−2.693	No	No	No
17	Linoleic Acid	−5.862	1.57	92.329	−2.723	No	No	No
18	Allixin	−3.074	1.301	93.438	−3.141	No	No	No
19	Ajoene	−3.54	1.329	95.186	−1.745	No	No	No
20	Ethyl Linoleate	−7.525	1.608	92.241	−2.774	No	No	Yes
21	Diazinon	−3.757	1.509	92.749	−3.005	No	No	No
22	Levamisole	−3.173	1.491	93.678	−2.075	No	No	No
23	Scutellarein	−3.156	−0.357	66.687	−2.735	Yes	No	No
24	S-allylcysteine methylester	−2.213	0.986	93.247	−3.061	No	No	No
25	Caffeic acid	−2.33	0.634	69.407	−2.722	No	No	No

**Table 5 biomedicines-11-00643-t005:** Distribution properties of ligands.

Sr. No	Ligand	VDss (Human)	Fraction Unbound (Human)	BBB Permeability (Human)	CNS Permeability
1	-L-Alanine	−0.553	0.462	−0.271	−3.472
2	Allicin	−0.045	0.577	0.506	−2.312
3	Diallyl Sulfide	0.202	0.552	0.69	−2.102
4	Diallyl Disulfide	0.211	0.518	0.78	−2.21
5	Diallyl Trisulfide	0.216	0.483	0.767	−2.309
6	Glutathione	−0.377	0.463	−1.085	−3.903
7	L-Cysteine	−0.486	0.49	−0.398	−3.476
8	S-Allyl-Mercapto- Glutathione	−1.517	0.588	−1.475	−4.217
9	Thiocysteine	−0.501	0.47	−0.376	−3.5
10	Gamma-Glutamyl- L-Cysteine	−0.203	0.495	−1.994	−4.159
11	Gamma-Glutamyl- S-Allylcysteine	−0.48	0.452	−1.124	−4.02
12	Kaempferol	1.274	0.178	−0.939	−2.228
13	Quercetin	1.559	0.206	−1.098	−3.065
14	Myricetin	1.317	0.238	−1.493	−3.709
15	Fructan	−0.276	0.499	−1.886	−4.815
16	Lauric Acid	−0.631	0.26	0.057	−2.034
17	Linoleic Acid	−0.587	0.054	−0.142	−1.6
18	Allixin	−0.008	0.479	0.193	−2.86
19	Ajoene	0.083	0.395	0.703	−2.178
20	Ethyl Linoleate	0.306	0.015	0.776	−1.562
21	Diazinon	−0.348	0.329	−0.438	−3.029
22	Levamisole	0.428	0.358	0.358	−2.011
23	Scutellarein	0.587	0.192	−1.398	−2.363
24	S-allylcysteine methylester	-0.396	0.434	−0.119	−2.911
25	Caffeic acid	-1.098	0.529	−0.647	−2.608

**Table 6 biomedicines-11-00643-t006:** Parameters measuring metabolism of ligands.

Sr. No	Ligands	CYP- 2D6 Substrate	CYP- 3A4 Substrate	CYP- 2D6 Inhibitor	CYP- 2C19 Inhibitor	CYP- 2C9 Inhibitor	CYP- 2D6 Inhibitor	CYP- 3A4 Inhibitor
1	-L-Alanine	NO	NO	NO	NO	NO	NO	NO
2	Allicin	NO	NO	NO	NO	NO	NO	NO
3	Diallyl Sulfide	NO	NO	NO	NO	NO	NO	NO
4	Diallyl Disulfide	NO	NO	NO	NO	NO	NO	NO
5	Diallyl Trisulfide	NO	NO	NO	NO	NO	NO	NO
6	Glutathione	NO	NO	NO	NO	NO	NO	NO
7	L-Cysteine	NO	NO	NO	NO	NO	NO	NO
8	S-Allyl-Mercapto- Glutathione	NO	NO	NO	NO	NO	NO	NO
9	Thiocysteine	NO	NO	NO	NO	NO	NO	NO
10	Gamma-Glutamyl- L-Cysteine	NO	NO	NO	NO	NO	NO	NO
11	Gamma-Glutamyl- S-Allylcysteine	NO	NO	NO	NO	NO	NO	NO
12	Kaempferol	NO	NO	YES	NO	NO	NO	NO
13	Quercetin	NO	NO	YES	NO	NO	NO	NO
14	Myricetin	NO	NO	YES	NO	NO	NO	NO
15	Fructan	NO	NO	NO	NO	NO	NO	NO
16	Lauric Acid	NO	NO	NO	NO	NO	NO	NO
17	Linoleic Acid	NO	YES	YES	NO	NO	NO	NO
18	Allixin	NO	NO	YES	NO	NO	NO	NO
19	Ajoene	NO	NO	NO	NO	NO	NO	NO
20	EthylLinoleate	NO	YES	YES	NO	NO	NO	NO
21	Diazinon	NO	NO	NO	NO	NO	NO	YES
22	Levamisole	NO	NO	YES	NO	NO	NO	NO
23	Scutellarein	NO	NO	YES	NO	NO	YES	NO
24	S-allylcysteine methylester	NO	NO	NO	NO	NO	NO	NO
25	Caffeic acid	NO	NO	NO	NO	NO	NO	NO

**Table 7 biomedicines-11-00643-t007:** Excretion properties of ligands.

Sr. No	Ligands	Total Clearance	Renal OCT2 Substrate
1	-L-Alanine	0.365	No
2	Allicin	0.714	No
3	DiallylSulfide	0.555	No
4	DiallylDisulfide	0.547	No
5	DiallylTrisulfide	0.446	No
6	Glutathione	0.308	No
7	L-Cysteine	0.53	No
8	S-Allyl-Mercapto- Glutathione	0.333	No
9	Thiocysteine	0.369	No
10	Gamma-Glutamyl- L-Cysteine	Gamma-Glutamyl- 0.159 L-Cysteine	No
11	Gamma-Glutamyl- S-Allylcysteine	- 0.3	No
12	Kaempferol	0.477	No
13	Quercetin	0.407	No
14	Myricetin	0.422	No
15	Fructan	1.516	No
16	Lauric Acid	1.623	No
17	Linoleic Acid	1.936	No
18	Allixin	0.419	No
19	Ajoene	0.538	No
20	Ethyl Linoleate	2.08	No
21	Diazinon	0.391	No
22	Levamisole	0.475	No
23	Scutellarein	0.47	No
24	S-allylcysteine Methylester	0.487	No
25	Caffeic acid	0.508	No

**Table 8 biomedicines-11-00643-t008:** Predicts toxicity of ligands.

Ligands	Max. Tolerated Dose (Human) mg/Kg	hERG I inhibitor	hERGII inhibitor	Oral Rat Acute Toxicity	Oral Rat Chronic Toxicity	Hepa toxicity	Skin Sensitization	t.Pyriformis Toxicity	Minnow Toxicity
-L-Alanine	1.164	No	No	2.051	1.9	No	No	0.268	2.598
Allicin	0.737	No	No	2.366	1.406	No	Yes	0.9	1.235
DiallylSulfide	0.782	No	No	2.028	1.812	No	Yes	0.63	1.154
DiallylDisulfide	0.674	No	No	2.375	1.847	No	Yes	1.371	0.79
DiallylTrisulfide	0.582	No	No	2.711	1.8 57	No	Yes	2.008	0.516
Glutathione	1.104	No	No	2.468	2.919	NO	NO	0.285	4.569
L-Cysteine	1.133	No	No	1.982	2.6	NO	NO	0.149	2.992
S-Allyl-Mercapto- Glutathione	1.196	No	No	1.804	2.902	YES	NO	0.285	4.569
Thiocysteine	1.113	No	No	1.983	2.275	NO	NO	0.149	2.992
Gamma-Glutamyl- L-Cysteine	0.856	No	No	2.478	3.361	YES	NO	0.285	4.164
Gamma-Glutamyl- S-Allylcysteine	1.119	No	No	2.438	3.361	NO	NO	0.101	2.657
Kaempferol	0.531	No	No	2.449	2.29	NO	NO	0.285	4.306
Quercetin	0.499	No	No	2.471	2.505	NO	NO	0.285	2.928
Myricetin	0.51	No	No	2.497	2.618	NO	NO	0.314	2.885
Fructan	0.667	No	No	2.775	2.718	NO	NO	0.288	3.721
Lauric Acid	−0.34	No	No	1.511	4.703	NO	NO	0.286	5.023
Linoleic Acid	−8.27	No	No	1.429	2.89	NO	NO	0.285	13.29
Allixin	−0.879	No	No	2.195	3.187	NO	NO	0.945	−0.084
Ajoene	0.462	No	No	2.472	0.899	NO	YES	0.701	−1.31
EthylLinoleate	0.009	No	No	1.644	3.023	NO	YES	0.324	1.582
Diazinon	1.362	No	No	3.258	0.953	YES	NO	2.197	0.155
Levamisole	0.035	No	No	2.711	1.548	NO	YES	1.497	−1.765
Scutellarein	0.626	No	No	2.452	3.135	NO	NO	0.366	−0.148
S-allylcysteine methylester	0.703	No	No	2.6	0.908	NO	NO	1.355	1.45
Caffeic acid	1.145	No	No	2.383	2.092	NO	NO	0.301	1.99

**Table 9 biomedicines-11-00643-t009:** Absorption properties of Remdesivir.

Ligands	Water Solubility	CaCO_2_ Permeability	Intestinal Absorption (human)	Skin Permeability	P-Gluco Protein Substrate	P-Gluco Protein I Inhibitor	P-Gluco Protein II Inhibitor
Remdesivir	−3.07	0.635	71.109	2.735	Yes	Yes	No

**Table 10 biomedicines-11-00643-t010:** Distribution properties of Remdesivir.

Ligand	VDss (Human)	Fraction Unbound (Human)	BBB Permeability (Human)	CNS Permeability
Remdesivir	0.307	0.005	−2.056	−4.675

**Table 11 biomedicines-11-00643-t011:** Metabolic properties of Remdesivir.

Ligand	CYP2D6 Substrate	CYP3A4 Substrate	CYP2D6 Inhibitor	CYP2C19 Inhibitor	CYP2C9 Inhibitor	CYP2D6 Inhibitor	CYP3A4 Inhibitor
Remdesivir	NO	YES	NO	NO	NO	NO	NO

**Table 12 biomedicines-11-00643-t012:** Excretion properties of Remdesivir.

Ligands	Total Clearance	Renal OCT2 Substrate
Remdesivir	0.198	NO

**Table 13 biomedicines-11-00643-t013:** Toxicity properties of Remdesivir.

Ligands	Max. Tolerated Dose (Human)	Herg I Inhibitor	Herg II Inhibitor	Oral Rat Acute Toxicity	Oral Rat Chronic Toxicity	Hepatoxicity	Skin Sensitization	T.Pyriformis Toxicity	Minnow Toxicity (Log Mm)
Remdesivir	1.972	No	Yes	2.043	1.639	Yes	No	0.285	0.291

**Table 14 biomedicines-11-00643-t014:** Docking result of Remdesivir.

Ligands	Binding Score	Cavity Size	Grid Map	HBA	HBD	logP	Mol. Weight g/mol
Remdesivir	−8.1	1385	22	13	4	2.31218	602.585

**Table 15 biomedicines-11-00643-t015:** Remdesivir comparison with the lead compound.

Ligand	LogP	Molecular Weight (g/mol)	Hydrogen Bond Acceptor	Hydrogen Bond Donar
Remdesivir	2.31218	602.585	13	4
Levamisole	2.1461	204.29	3	0

**Table 16 biomedicines-11-00643-t016:** Comparative values of absorption of Remdesivir and Levamisole.

Ligands	Water Solubility	CaCO_2_ Permeability	Intestinal Absorption (Human)	Skin Permeability	P-Glucoprotein Substrate	P-Glucoprotein I Inhibitor	PglucoproteinII Inhibitor
Remdesivir	−3.07	0.635	71.109	−2.735	Yes	Yes	No
Levamisole	−3.173	1.491	93.678	−2.075	No	No	No

**Table 17 biomedicines-11-00643-t017:** Comparative values of metabolic properties of Remdesivir and Levamisole.

Ligand	CYP2D6 Substrate	CYP3A4 Substrate	CYP2D6 Inhibitor	CYP2C19 Inhibitor	CYP2C9 Inhibitor	CYP2D6 Inhibitor	CYP2D6 Inhibitor
Remdesivir	No	Yes	No	No	No	No	No
Levamisole	No	No	Yes	No	No	Yes	No

**Table 18 biomedicines-11-00643-t018:** Comparative values of the distribution of Remdesivir and Levamisole.

Ligands	VDss (Human)	Fraction Unbound (Human)	BBB Permeability (Human)	CNS Permeability
Remdesivir	0.307	0.005	−2.056	−4.675
Levamisole	0.428	0.358	0.358	−2.011

**Table 19 biomedicines-11-00643-t019:** Values of excretory properties of Remdesivir and Levamisole.

Ligands	Total Clearance	Renal OCT2 Substrate
Remdesivir	0.198	No
Levamisole	0.475	No

**Table 20 biomedicines-11-00643-t020:** Comparative values of toxicity of Remdesivir and Levamisole.

Ligand	Max. Tolerated Dose (Human) (mg/Kg)	HergI Inhibitor	HergII Inhibitor	Oral Rat Acute Toxicity (mol/Kg)	Oral Rat Chronic Toxicity (mol/Kg)	Hepa Toxicity	Skin Sensitization	*T.Pyriforms* Toxicity (LogUg/L)	Minnow Toxicity (Log Mm)
Remdesivir	0.196	No	Yes	2.043	1.639	Yes	No	0.285	0.291
Levamisole	0.035	No	No	2.711	1.548	No	Yes	1.355	1.45

**Table 21 biomedicines-11-00643-t021:** Comparison of physiochemical properties of Remdesivir and Levamisole.

Ligands	LogP	Molecular Weight (g/mol)	Molecular Formula	HBond Acceptor	HBond Donar
Remdesivir	2.31218	602.585	C27H35N6O8P	13	4
Levamisole	2.1461	204.29	C11H12N2S	3	0

**Table 22 biomedicines-11-00643-t022:** Docking Score Comparison of Levamisole and Remdesivir.

Ligands	Score
Remdesivir	−8.1
Levamisole	−5.7

## Data Availability

All data generated or analyzed during this study are included in this article.

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
