# Peer review of "Molecular Screening of Bioactive Compounds of Garlic for Therapeutic Effects against COVID-19"

_biomedicines, 2023, doi:10.3390/biomedicines11020643_

Round 1
Reviewer 1 Report
The manuscript entitled “ Immuno-regulatory and Anti-Inflammatory Effects of Bioactive Compounds of Garlic against Covid-19“ is very interesting and manuscript is well written with excellent figure included, therefore, although very informative, it is easy to read the manuscript in its present form.
The statistical methods are clearly presented ant the organization of the article follows the standard recommendations of an original article and English language is acceptable, but needs small improvement.
There are certain points that could be addressed and improve the quality of this manuscript:
- Please , add objectives of your research in the introduction
- Allium sativum should be written in italicum form
- Make the list of abbreviations at the end of the manuscript
-„Medicinal plants are those plants in which healing properties are present and they show detrimental effects medicinal effects on the human or animal body.“ – this sentence is not written correctly and it seems confusing.
-„But the distribution of medicinal plants is not the same worldwide and usually, medicinal herbs are collected from wildlife populations.“ – The sentence should not star with conjunction, so please change that.
- Authors should include critical analysis with discussion of strengths and weaknesses of important studies that are cited.
Author Response
Dear Editor,
We are pleased to submit the revised version of the manuscript and have addressed all the comments raised by the reviewer.
We hope that a revised version of the manuscript will be considered by Biomedicine. We have modified the paper in light of extensive and insightful reviewers’ comments. The response to the editor’s and reviewer reforming comments is given below. Changes have been highlighted in yellow in the manuscript.
Response to Reviewer’s comments:
Reviewer1
1.- Please, add objectives of your research in the introduction
Answer: Objectives have been added in 2nd last paragraph of page 2.
- - Allium sativum should be written in italicum form
Answer: Allium sativum has been written in italicum form throughout the manuscript.
- - Make the list of abbreviations at the end of the manuscript
Answer: Abbreviations have been added at the end of the article.
- -„Medicinal plants are those plants in which healing properties are present and they show detrimental effects medicinal effects on the human or animal body.“ – this sentence is not written correctly and it seems confusing.
Answer: During the language editing this sentence has been removed.
5, -„But the distribution of medicinal plants is not the same worldwide and usually, medicinal herbs are collected from wildlife populations.“ – The sentence should not star with conjunction, so please change that.
Answer: Sentence has been repharased in the introduction section as "The problem is that the distribution of medicinal plants is not same worldwide and usually, medicinal herbs are collected from wildlife populations” (2nd page 3rd paragraph).
- - Authors should include critical analysis with a discussion of the strengths and weaknesses of important studies that are cited.
Answer: Discussion has been added in the discussion section.
Reviewer 2 Report
The manuscript entitled "Immuno-regulatory and Anti-Inflammatory Effects of Bioactive Compounds of Garlic against Covid-19" describes the ADME properties and molecular docking of active constituents of Garlic.
The manuscript is lacking a rationale. The authors just carried out all possible computational studies of the active constituents of Garlic without any further details discussions or evidence.
It is very unclear how these can be used as drug candidates for COVID-19. Also, the title of the manuscript is not justified with the content of the manuscript.
There is no discussion of how Remdesivir is correlated with these active constituents.
The manuscript is very poorly written and has numerous grammatical and spelling errors.
Author Response
Dear Editor,
We are pleased to submit the revised version of the manuscript and have addressed all the comments raised by the reviewer.
We hope that a revised version of the manuscript will be considered by Biomedicine. We have modified the paper in light of extensive and insightful reviewers’ comments. The response to the editor’s and reviewers reforming comments is given below. Changes have been highlighted in yellow in the manuscript.
Response to Reviewer’s comments:
Reviewer2
The manuscript entitled "Immuno-regulatory and Anti-Inflammatory Effects of Bioactive Compounds of Garlic against Covid-19" describes the ADME properties and molecular docking of active constituents of Garlic.
The manuscript is lacking a rationale. The authors just carried out all possible computational studies of the active constituents of Garlic without any further details discussions or evidence.
Answer: Discussion is added
It is very unclear how these can be used as drug candidates for COVID-19. Also, the title of the manuscript is not justified with the content of the manuscript.
Answer: Title has been modified accordingly.
There is no discussion of how Remdesivir is correlated with these active constituents.
Answer: Discussion is added. Remdesivir is the only FDA-approved drug against covid 19- so a comparison was made regarding all parameters of absorption, distribution, metabolism, excretion, and toxicity.
The manuscript is very poorly written and has numerous grammatical and spelling errors.
Answer: Language has been improved.
Reviewer 3 Report
The article “Immuno-regulatory and Anti-Inflammatory Effects of Bioactive Compounds of Garlic against COVID-19” is devoted to an important topic. I recommend this paper to be published in the journal. Here are some major suggestions:
1: In “Title”, “Covid-19” should corrected to “COVID-19”.
2: Please check references carefully.
3: The “Abstract” and the body of the manuscript should be more concise. Please make the necessary correction.
4: In “Introduction”, it is suggested to add some background and highlight the novelty of this work clearly. “Many drugs are available in the market against SARS-CoV-2 but one cannot say which one is perfect from all the above drugs” Please extend this sentence and cite relevant references. For example, “Effective measures, such as vaccines (Nature Immunol. 2022, 23, 360-370), small-molecule inhibitors (J. Med. Virol. 2022, 94, 1373-1390; Pharmaceuticals. 2022, 15, 165.), and bioactive natural products (Biomedicines. 2021, 9, 689; Viruses. 2021, 13, 609) are greatly needed to reduce SARS-CoV-2 transmission. However, promising drugs for SARS-CoV-2 treatment do not exist (Front. Immunol. 2022, 13, 1015355). As a key component of the COVID-19 treatment regimen, transactional medicine (Pharmaceutics. 2021, 13, 1839; J. Ethnopharmacol. 2021, 270, 113869), including Garlic, may demonstrate potential value in countering SARS-CoV-2 infection.” This is critical to address in this manuscript, the authors should enrich this part in the revised version.
5: In “Introduction”, “These spreader viruses are grouped into, alpha, beta, gamma and delta variants.” Please add the current Omicron variant.
6: “By July 2021, globally 100 million confirmed cases of COVID-19 were reported with more than 2 million deaths.” Please update relevant data.
7: Some unnecessary “Tables” or “Figures” should be removed in the revised version.
Author Response
Dear Editor,
We are pleased to submit the revised version of the manuscript and have addressed all the comments raised by reviewer.
We hope that a revised version of the manuscript will be considered by Biomedicine. We have modified the paper in light of extensive and insightful reviewers’ comments. The response to the editor’s and reviewer reforming comments is given below. Changes have been highlighted in yellow in the manuscript.
Response to Reviewer’s comments:
Reviewer3
The article “Immuno-regulatory and Anti-Inflammatory Effects of Bioactive Compounds of Garlic against COVID-19” is devoted to an important topic. I recommend this paper to be published in the journal. Here are some major suggestions:
1: In “Title”, “Covid-19” should correct to “COVID-19”.
Answer: Correction done throughout the manuscript.
2: Please check references carefully.
Answer: All the references have been rechecked and updated accordingly.
3: The “Abstract” and the body of the manuscript should be more concise. Please make the necessary correction.
Answer: The abstract and the body of the manuscript has been restructured and have been made more concise.
4: In the “Introduction”, it is suggested to add some background and highlight the novelty of this work clearly. “Many drugs are available in the market against SARS-CoV-2 but one cannot say which one is perfect from all the above drugs” Please extend this sentence and cite relevant references. For example, “Effective measures, such as vaccines (Nature Immunol. 2022, 23, 360-370), small-molecule inhibitors (J. Med. Virol. 2022, 94, 1373-1390; Pharmaceuticals. 2022, 15, 165.), and bioactive natural products (Biomedicines. 2021, 9, 689; Viruses. 2021, 13, 609) are greatly needed to reduce SARS-CoV-2 transmission. However, promising drugs for SARS-CoV-2 treatment do not exist (Front. Immunol. 2022, 13, 1015355). As a key component of the COVID-19 treatment regimen, transactional medicine (Pharmaceutics. 2021, 13, 1839; J. Ethnopharmacol. 2021, 270, 113869), including Garlic, may demonstrate potential value in countering SARS-CoV-2 infection.” This is critical to address in this manuscript, the authors should enrich this part in the revised version.
Answer: The information has been rephrased with proper references as suggested (2nd page, 3rd paragraph).
5: In “Introduction”, “These spreader viruses are grouped into, alpha, beta, gamma and delta variants.” Please add the current Omicron variant.
Answer: information is updated in the introduction.
6: “By July 2021, globally 100 million confirmed cases of COVID-19 were reported with more than 2 million deaths.” Please update relevant data.
Answer: Data is updated in the introduction with proper citation.
7: Some unnecessary “Tables” or “Figures” should be removed in the revised version.
Answer: Unnecessary tables and figures have been removed.
Round 2
Reviewer 2 Report
The authors revised the manuscript as suggested.
Reviewer 3 Report
The authors have addressed my comments to my satisfaction. I recommend it for publication in its current form.